# Simultaneous Rectification and Alignment via Robust Recovery of Low-rank Tensors

**Xiaoqin Zhang, Di Wang**
Institute of Intelligent System and Decision
Wenzhou University
zhangxiaoqinnan@gmail.com, wangdi@wzu.edu.cn

**Zhengyuan Zhou**
Department of Electrical Engineering
Stanford University
zyzhou@stanford.edu

**Yi Ma**
Visual computing group
Microsoft Research Asia
mayi@microsoft.com

## Abstract

In this work, we propose a general method for recovering low-rank three-order tensors, in which the data can be deformed by some unknown transformation and corrupted by arbitrary sparse errors. Since the unfolding matrices of a tensor are interdependent, we introduce auxiliary variables and relax the hard equality constraints by the augmented Lagrange multiplier method. To improve the computational efficiency, we introduce a proximal gradient step to the alternating direction minimization method. We have provided proof for the convergence of the linearized version of the problem which is the inner loop of the overall algorithm. Both simulations and experiments show that our methods are more efficient and effective than previous work. The proposed method can be easily applied to simultaneously rectify and align multiple images or videos frames. In this context, the state-of-the-art algorithms "RASL" and "TILT" can be viewed as two special cases of our work, and yet each only performs part of the function of our method.

## 1   Introduction

In recent years, with the advances in sensorial and information technology, massive amounts of high-dimensional data are available to us. It has become an increasingly pressing challenge to develop efficient and effective computational tools that can automatically extract the hidden structures and hence useful information from such data. Many revolutionary new tools have been developed that enable people to recover low-dimensional structures in the form of sparse vectors or low-rank matrices in high dimensional data. Nevertheless, instead of vectors and matrices, many practical data are given in their natural form as higher-order tensors, such as videos, hyper-spectral images, and 3D range data. These data are often subject to all types of geometric deformation or corruptions due to change of viewpoints, illuminations or occlusions. The true intrinsic structures of the data will not be fully revealed unless these nuisance factors are undone in the processing stage.

In the literature, it has been shown that for matrix data, if the data is a deformed or corrupted version of an intrinsically low-rank matrix, one can recover the rectified low-rank structure despite different types of deformation (linear or nonlinear) and severe corruptions. Such concepts and methods have been successfully applied to rectify the so-called low-rank textures [1] and to align multiple correlated images (such as video frames or human faces) [2, 3, 4, 5, 6]. However, when applied to the data of higher-order tensorial form, such as videos or 3D range data, these tools are only able to harness one type of low-dimensional structure at a time, and are not able to exploit the low-dimensional

tensorial structures in the data. For instance, the previous work of TILT rectifies a low-rank textural region in a single image [1] while RASL aligns multiple correlated images [6]. They are highly complementary to each other: they exploit spatial and temporal linear correlations respectively in a given sequence of images. A natural question arises: can we simultaneously harness all such low-dimensional structures in an image sequence by viewing it as a three-order tensor?

Actually, many existing visual data can be naturally viewed as three-order (or even higher-order) tensors (e.g. color images, videos, hyper-spectral images, high-dynamical range images, 3D range data etc.). Important structures or useful information will very often be lost if we process them as a 1D signal or a 2D matrix. For tensorial data, however, one major challenge lies in an appropriate definition of the rank of a tensor, which corresponds to the notion of intrinsic "dimension" or "degree of freedom" for the tensorial data. Traditionally, there are two definitions of tensor rank, which are based on PARAFAC decomposition [7] and Tucker decomposition [8] respectively. Similar to the definition of matrix rank, the rank of a tensor based on PARAFAC decomposition is defined as the minimum number of rank-one decompositions of a given tensor. However, this definition rank is a nonconvex and nonsmooth function on the tensor space, and direct minimization of this function is an NP-hard problem. An alternative definition of tensor rank is based on the so-called Tucker decomposition, which results in a vector of the ranks of a set of matrices unfolded from the tensor.

Due to the recent breakthroughs in the recovery of low-rank matrices [9], the latter definition has received increasing attention. Gandy *et al.* [10] adopt the sum of the ranks of the different unfolding matrices as the rank of the tensor data, which is in turn approximated by the sum of their nuclear norms. They then apply the alternating direction method (ADM) to solve the tensor completion problem with Gaussian observation noise. Instead of directly adding up the ranks of the unfolding matrices, a weighted sum of the ranks of the unfolding matrices is introduced by Liu *et al.* [12] and they also proposed several optimization algorithms to estimate missing values for tensorial visual data (such as color images). In [13], three different strategies have been developed to extend the trace-norm regularization to tensors: (1) tensors treated as matrices; (2) traditional constrained optimization of low rank tensors as in [12]; (3) a mixture of low-rank tensors. The above-mentioned work all addresses the tensor completion problem in which the locations of the missing entries are known, and moreover, observation noise is assumed to be Gaussian. However, in practice, a fraction of the tensorial entries can be arbitrarily corrupted by some large errors, and the number and the locations of the corrupted entries are unknown. Li *et al.* [14] have extended the Robust Principal Component Analysis [9] from recovering a low-rank matrix to the tensor case. More precisely, they have proposed a method to recover a low-rank tensor with sparse errors. However, there are two issues that limit the practicality of such methods: (1) The tensorial data are assumed to be well aligned and rectified. (2) The optimization method can be improved in both accuracy and efficiency, which will be discussed and validated in Section 4.

Inspired by the previous work and motivated by the above observations, we propose a more general method for the recovery of low-rank tensorial data, especially three-order tensorial data, since our main interests are visual data. The main contributions of our work are three-fold: (1) The data samples in the tensor do not need to be well-aligned or rectified, and can be arbitrarily corrupted with a small fraction of errors. (2) This framework can simultaneously perform rectification and alignment when applied to imagery data such as image sequences and video frames. In particular, existing work of RASL and TILT can be viewed as two special cases of our method. (3) To resolve the interdependence among the nuclear norms of the unfolding matrices, we introduce auxiliary variables and relax the hard equality constraints using the augmented Lagrange multiplier method. To further improve the efficiency, we introduce a proximal gradient step to the alternating direction minimization method. The optimization is more efficient and effective than the previous work [6, 14], and the convergence (of the linearized version) is guaranteed (the proof is shown in the supplementary material).

## 2   Basic Tensor Algebra

We provide a brief notational summary here. Lowercase letters $(a, b, c \cdots)$ denote scalars; bold lowercase $(\boldsymbol{a}, \boldsymbol{b}, \boldsymbol{c} \cdots)$ letters denote vectors; capital letters $(A, B, C \cdots)$ denote matrices; calligraphic letters $(\mathcal{A}, \mathcal{B}, \mathcal{C} \cdots)$ denote tensors. In the following subsections, the tensor algebra and the tensor rank are briefly introduced.

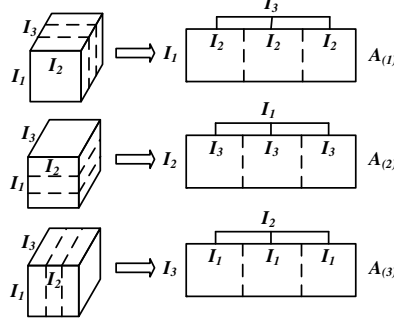

Figure 1: Illustration of unfolding a 3-order tensor.

## 2.1 Tensor Algebra

We denote an $N$-order tensor as $\mathcal{A} \in \mathbb{R}^{I_1 \times I_2 \times \cdots \times I_N}$, where $I_n(n = 1, 2, \ldots, N)$ is a positive integer. Each element in this tensor is represented as $a_{i_1 \cdots i_n \cdots i_N}$, where $1 \leq i_n \leq I_n$. Each order of a tensor is associated with a 'mode'. By unfolding a tensor along a mode, a tensor's unfolding matrix corresponding to this mode is obtained. For example, the mode-$n$ unfolding matrix $A_{(n)} \in \mathbb{R}^{I_n \times (\prod_{i \neq n} I_i)}$ of $\mathcal{A}$, represented as $A_{(n)} = \mathrm{unfold}_n(\mathcal{A})$, consists of $I_n$-dimensional mode-$n$ column vectors which are obtained by varying the $n$th-mode index $i_n$ and keeping indices of the other modes fixed. Fig. 1 shows an illustration of unfolding a 3-order tensor. The inverse operation of the mode-$n$ unfolding is the mode-$n$ folding which restores the original tensor $\mathcal{A}$ from the mode-$n$ unfolding matrix $A_{(n)}$, represented as $\mathcal{A} = \mathrm{fold}_n(A_{(n)})$. The mode-$n$ rank $r_n$ of $\mathcal{A}$ is defined as the rank of the mode-$n$ unfolding matrix $A_{(n)}$: $r_n = \mathrm{rank}(A_{(n)})$. The operation of mode-$n$ product of a tensor and a matrix forms a new tensor. The mode-$n$ product of tensor $\mathcal{A}$ and matrix $U$ is denoted as $\mathcal{A} \times_n U$. Let matrix $U \in \mathbb{R}^{J_n \times I_n}$. Then, $\mathcal{A} \times_n U \in \mathbb{R}^{I_1 \times \cdots \times I_{n-1} \times J_n \times I_{n+1} \times \cdots \times I_N}$ and its elements are calculated by:

$$(\mathcal{A} \times_n U)_{i_1 \cdots i_{n-1} j_n i_{n+1} \cdots i_N} = \sum_{i_n} a_{i_1 \cdots i_n \cdots i_N} u_{j_n i_n}. \tag{1}$$

The scalar product of two tensors $\mathcal{A}$ and $\mathcal{B}$ with the dimension is defined as $\langle \mathcal{A}, \mathcal{B} \rangle = \sum_{i_1} \sum_{i_2} \cdots \sum_{i_N} a_{i_1 \cdots i_N} b_{i_1 \cdots i_N}$. The Frobenius norm of $\mathcal{A} \in \mathbb{R}^{I_1 \times I_2 \times \cdots \times I_N}$ is defined as: $||\mathcal{A}||_F = \sqrt{\langle \mathcal{A}, \mathcal{A} \rangle}$. The $l_0$ norm $||\mathcal{A}||_0$ is defined to be the number of non-zero entries in $\mathcal{A}$ and the $l_1$ norm $||\mathcal{A}||_1 = \sum_{i_1, \ldots, i_N} |a_{i_1 \cdots i_N}|$ respectively. Observe that $||\mathcal{A}||_F = ||A_{(k)}||_F$, $||\mathcal{A}||_0 = ||A_{(k)}||_0$ and $||\mathcal{A}||_1 = ||A_{(k)}||_1$ for any $1 \leq k \leq N$.

## 2.2 Tensor Rank

Traditionally, there are two definitions of tensor rank, which are based on PARAFAC decomposition [7] and Tucker decomposition [8], respectively.

As stated in [7], in analogy to SVD, the rank of a tensor $\mathcal{A}$ can be defined as the minimum number $r$ for decomposing the tensor into rank-one components as follows:

$$\mathcal{A} = \sum_{j=1}^{r} \lambda_j \boldsymbol{u}_j^{(1)} \circ \boldsymbol{u}_j^{(2)} \circ \cdots \circ \boldsymbol{u}_j^{(N)} = \mathcal{D} \times_1 U^{(1)} \times_2 U^{(2)} \cdots \times_N U^{(N)}, \tag{2}$$

where $\circ$ denotes outer product, $\mathcal{D} \in \mathbb{R}^{r \times r \times \cdots \times r}$ is an $N$-order diagonal tensor whose $j$th diagonal element is $\lambda_j$, and $U^{(n)} = [\boldsymbol{u}_1^{(n)}, \ldots, \boldsymbol{u}_r^{(n)}]$. The above decomposition model is called PARAFAC. However, this rank definition is a highly nonconvex and discontinuous function on the tensor space. In general, direct minimization of such a function is NP-hard.

Another kind of rank definition considers the mode-$n$ rank $r_n$ of tensors, which is inspired by the Tucker decomposition [8]. The tensor $\mathcal{A}$ can be decomposed as follows:

$$\mathcal{A} = \mathcal{G} \times_1 U^{(1)} \times_2 U^{(2)} \cdots \times_N U^{(N)}, \tag{3}$$

where $\mathcal{G} = \mathcal{A} \times_1 U^{(1)^\top} \times_2 U^{(2)^\top} \cdots \times_N U^{(N)^\top}$ is the core tensor controlling the interaction between the $N$ mode matrices $U^{(1)}, \ldots, U^{(N)}$. In the sense of Tucker decomposition, an appropriate definition of tensor rank should satisfy the follow condition: a low-rank tensor is a low-rank matrix when unfolded appropriately. This means the rank of a tensor can be represented by the rank of the

tensor's unfolding matrices. As illustrated in [8], the orthonormal column vectors of $U^{(n)}$ span the column space of the mode-$n$ unfolding matrix $A_{(n)}, (1 \leq n \leq N)$, so that if $U^{(n)} \in \mathcal{R}^{I_n \times r_n}, n = 1, \ldots, N$, then the rank of the mode-$n$ unfolding matrix $A_{(n)}$ is $r_n$. Accordingly, we call $\mathcal{A}$ a rank-$(r_1, \ldots, r_N)$ tensor. We adopt this tensor rank definition in this paper.

## 3 Low-rank Structure Recovery for Tensors

In this section, we first formulate the problem of recovering low-rank tensors despite deformation and corruption, and then introduce an iterative optimization method to solve the low-rank recovery problem. Finally, the relationship between our work and the previous work is discussed to show why our work can simultaneously realize rectification and alignment.

### 3.1 Problem Formulation

Without loss of generality, in this paper we focus on 3-order tensors to study the low-rank recovery problem. Most practical data and applications we experiment with belong to this class of tensors. Consider a low-rank 3-order data tensor $\mathcal{A} \in \mathbb{R}^{I_1 \times I_2 \times I_3}$. In real applications, the data are inevitably corrupted by noise or errors. Rather than modeling the noise with a small Gaussian, we model it with an additive sparse error term $\mathcal{E}$ which fulfills the following conditions: (1) only a small fraction of entries are corrupted; (2) the errors are large in magnitude; (3) the number and the location of the corrupted data are unknown.

Based on the above assumptions, the original tensor data $\mathcal{A}$ can be represented as

$$\mathcal{A} = \mathcal{L} + \mathcal{E}, \tag{4}$$

where $\mathcal{L}$ is a low-rank tensor. In this paper, the notion of low-rankness will become clear once we introduce our objective function in a few paragraphs. The ultimate goal of this work is to recover $\mathcal{L}$ from the erroneous observations $\mathcal{A}$.

An explicit assumption in Eq. (4) is that it requires the tensor to be well aligned. For real data such as video and face images, the image frames (face images) should be well aligned to ensure that the three-order tensor of the image stack to have low-rank. However, for most practical data, precise alignments are not always guaranteed and even small misalignments will break the low-rank structure of the data. To compensate for possible misalignments, we adopt a set of transformations $\boldsymbol{\tau}_1^{-1}, \ldots, \boldsymbol{\tau}_{I_3}^{-1} \in \mathbb{R}^p$ ($p$ is the dimension of the transformations) which act on the two-dimensional slices (matrices) of the tensor data[1]. Based on the set of transformations $\Gamma = \{\boldsymbol{\tau}_1, \ldots, \boldsymbol{\tau}_{I_3}\}$, Eq. (4) can be changed to

$$\mathcal{A} \circ \Gamma = \mathcal{L} + \mathcal{E}, \tag{5}$$

where $\mathcal{A} \circ \Gamma$ means applying the transformation $\boldsymbol{\tau}_i$ to each matrix $\mathcal{A}(:, :, i), i = 1, \ldots, I_3$.

When both corruption and misalignment are modeled, the low-rank structure recovery for tensors can be formalized as follows.

$$\min_{\mathcal{L}, \mathcal{E}, \boldsymbol{\Gamma}} \quad \text{rank}(\mathcal{L}) + \gamma ||\mathcal{E}||_0, \quad \text{s.t.} \quad \mathcal{A} \circ \Gamma = \mathcal{L} + \mathcal{E}. \tag{6}$$

The above optimization problem is not directly tractable for the following two reasons: (1) both rank and $\ell_0$-norm are nonconvex and discontinuous; (2) the equality constraint $\mathcal{A} \circ \Gamma = \mathcal{L} + \mathcal{E}$ is highly nonlinear due to the domain transformation $\Gamma$.

To relax the limitation (1), we first recall the tensor rank definition in Section 2.2. In our work, we adopt the rank definition based on the Tucker decomposition which can be represented as follows: $\mathcal{L}$ is a rank-$(r_1, r_2, r_3)$ tensor where $r_i$ is the rank of unfolding matrix $L_{(i)}$. In this way, tensor rank can be converted to calculating a set of matrices' rank. We know that the nuclear (or trace) norm is the convex envelop of the rank of matrix: $||L_{(i)}||_* = \sum_{k=1}^m \sigma_k(L_{(i)})$, where $\sigma_k(L_{(i)})$ is $k$th singular value of matrix $L_{(i)}$. Therefore, we define the nuclear norm of a three-order tensor as follows:

$$||\mathcal{L}||_* = \sum_{i=1}^N \alpha_i ||L_{(i)}||_*, \ N = 3. \tag{7}$$

We assume $\sum_{i=1}^N \alpha_i = 1$ to make the definition consistent with the form of matrix. The rank of $\mathcal{L}$ is replaced by $||\mathcal{L}||_*$ to make a convex relaxation of the optimization problem. It is well know that

$\ell_1$-norm is a good convex surrogate of the $\ell_0$-norm. We hence replace the $||\mathcal{E}||_0$ with $||\mathcal{E}||_1$ and the optimization problem in (6) becomes

$$\min_{\mathcal{L},\mathcal{E},\mathbf{\Gamma}} \sum_{i=1}^{3} \alpha_i||L_{(i)}||_* + \gamma||\mathcal{E}||_1, \quad \text{s.t.} \quad \mathcal{A} \circ \Gamma = \mathcal{L} + \mathcal{E}. \tag{8}$$

For limitation (2), linearization with respect to the transformation $\Gamma$ parameters is a popular way to approximate the above constraint when the change in $\boldsymbol{\tau}$ is small or incremental. Accordingly, the first-order approximation to the above problem is as follows.

$$\min_{\mathcal{L},\mathcal{E},\Delta\Gamma} \sum_{i=1}^{3} \alpha_i||L_{(i)}||_* + \gamma||\mathcal{E}||_1, \quad \text{s.t.} \quad \mathcal{A} \circ \Gamma + \text{fold}_3\left((\sum_{i=1}^{n} J_i \Delta\Gamma \boldsymbol{\epsilon}_i \boldsymbol{\epsilon}_i^\top)^\top\right) = \mathcal{L} + \mathcal{E}, \tag{9}$$

where $J_i$ represents the Jacobian of $\mathcal{A}(:,:,i)$ with respect to the transformation parameters $\boldsymbol{\tau}_i$, and $\boldsymbol{\epsilon}_i$ denotes the standard basis for $\mathbb{R}^n$.

## 3.2 Optimization Algorithm

Although the problem in (9) is convex, it is still difficult to solve due to the interdependent nuclear norm terms. To remove these interdependencies and to optimize these terms independently, we introduce three auxiliary matrices $\{M_i, i = 1, 2, 3\}$ to replace $\{L_{(i)}, i = 1, 2, 3\}$, and the optimization problem changes to

$$\min_{\mathcal{L},\mathcal{E},\Delta\tilde{\Gamma}} \sum_{i=1}^{3} \alpha_i||M_i||_* + \gamma||\mathcal{E}||_1 \quad \text{s.t.} \quad \mathcal{A} \circ \Gamma + \Delta\tilde{\Gamma} = \mathcal{L} + \mathcal{E}, L_{(i)} = M_i, \ i = 1, 2, 3, \tag{10}$$

where we define $\Delta\tilde{\Gamma} \doteq \text{fold}_3\left((\sum_{i=1}^{n} J_i \Delta\Gamma \boldsymbol{\epsilon}_i \boldsymbol{\epsilon}_i^\top)^\top\right)$ for simplicity.

To relax the above equality constraints, we apply the Augmented Lagrange Multiplier (ALM) method [15] to the above problem, and obtain the following augmented Lagrangian function

$$
\begin{aligned}
f_\mu(M_i, \mathcal{L}, \mathcal{E}, \Delta\tilde{\Gamma}, \mathcal{Y}, Q_i) \quad = \quad & \sum_{i=1}^{3} \alpha_i||M_i||_* + \gamma||\mathcal{E}||_1 - \langle \mathcal{Y}, \mathcal{T} \rangle + \frac{1}{2\mu}||\mathcal{T}||_F^2 \\
& + \sum_{i=1}^{3} (-\langle Q_i, O_i \rangle + \frac{1}{2\mu_i}||O_i||_F^2),
\end{aligned}
\tag{11}
$$

where we define $\mathcal{T} = \mathcal{L} + \mathcal{E} - \mathcal{A} \circ \Gamma - \Delta\tilde{\Gamma}$ and $O_i = L_{(i)} - M_i$. $\mathcal{Y}$ and $Q_i$ are the Lagrange multiplier tensor and matrix respectively. $\langle \cdot, \cdot \rangle$ denotes the inner product of matrices or tensors. $\mu$ and $\mu_i$ are positive scalars. To have fewer parameters, we set $\mu = \mu_i$, $i = 1, 2, 3$ and $\mu_i$ is replaced by $\mu$ in the following sections including experiments and supplementary materials.

A typical iterative minimization process based on the alternating direction method of multipliers (ADMM) [15, 16] can be written explicitly as

$$
\begin{cases}
[M_i^{k+1}, \mathcal{L}^{k+1}, \mathcal{E}^{k+1}] : \quad & = \quad \arg\min_{M_i, \mathcal{L}, \mathcal{E}} f_\mu(M_i, \mathcal{L}, \mathcal{E}, \Delta\tilde{\Gamma}^k, \mathcal{Y}^k, Q_i^k); \\
\Delta\tilde{\Gamma}^{k+1} : \quad & = \quad \arg\min_{\Delta\tilde{\Gamma}} f_\mu(M_i^{k+1}, \mathcal{L}^{k+1}, \mathcal{E}^{k+1}, \Delta\tilde{\Gamma}, \mathcal{Y}^k, Q_i^k); \\
\mathcal{Y}^{k+1} : \quad & = \quad \mathcal{Y}^k - \mathcal{T}^{k+1}/\mu; \\
Q_i^{k+1} : \quad & = \quad Q_i^k - (\mathcal{L}_{(i)}^{k+1} - M_i^{k+1})/\mu, \quad i = 1, 2, 3.
\end{cases}
\tag{12}
$$

However, minimizing the augmented Lagrangian function $f_\mu(M_i, \mathcal{L}, \mathcal{E}, \Delta\tilde{\Gamma}^k, \mathcal{Y}^k, Q_i^k)$ with respect to $M_i$, $\mathcal{L}$ and $\mathcal{E}$ using ADMM is expensive in practice, and moreover, the global convergence can not be guaranteed. Therefore, we propose to solve the above problem by taking one proximal gradient step.

$$
\begin{cases}
M_i^{k+1} : \ = \arg\min_{M_i} \alpha_i||M_i||_* + \frac{1}{2\mu\tau_1} \left\| M_i - \left( M_i^k - \tau_1(M_i^k - \mathcal{L}_{(i)}^k + \mu Q_i^k) \right) \right\|_F^2, \ i = 1, 2, 3; \\[2mm]
\mathcal{L}^{k+1} : \ = \arg\min_{\mathcal{L}} \frac{1}{2\mu\tau_1} \left\| \mathcal{L} - \left( \mathcal{L}^k - \tau_1 \left( \sum_{i=1}^{3} \left( \mathcal{L}^k - \text{fold}_i(M_i^k + \mu Q_i^k) \right) + \mathcal{T}^k - \mu\mathcal{Y}^k \right) \right) \right\|_F^2; \\[2mm]
\mathcal{E}^{k+1} : \ = \arg\min_{\mathcal{E}} \gamma||\mathcal{E}||_1 + \frac{1}{2\mu\tau_1} \left\| \mathcal{E} - \left( \mathcal{E}^k - \tau_1 \left( \mathcal{T}^k - \mu\mathcal{Y}^k \right) \right) \right\|_F^2; \\[2mm]
\Delta\tilde{\Gamma}^{k+1} : \ = \arg\min_{\Delta\tilde{\Gamma}} \frac{1}{2\mu\tau_2} \left\| \Delta\tilde{\Gamma} - \left( \Delta\tilde{\Gamma}^k - \tau_2 \left( \Delta\tilde{\Gamma}^k - \mathcal{L}^{k+1} - \mathcal{E}^{k+1} + \mathcal{A} \circ \Gamma + \mu\mathcal{Y}^k \right) \right) \right\|_F^2.
\end{cases}
\tag{13}
$$

In detail, the solutions of each term are obtained as follows.

- For term $M_i^{k+1}$:
$$M_i^{k+1} = U_i D_{\alpha_i \mu \tau_1}(\Lambda) V_i^\top,$$
where $U_i \Lambda V_i^\top = M_i^k - \tau_1 (M_i^k - \mathcal{L}_{(i)}^k + \mu Q_i^k)$ and $D_\lambda(\cdot)$ is the shrinkage operator: $D_\lambda(x) = \operatorname{sgn}(x) \max(|x| - \lambda, 0)$.

- For term $\mathcal{L}^{k+1}$:
$$\mathcal{L}^{k+1} = \mathcal{L}^k - \tau_1 \Big( \sum_{i=1}^3 \left( \mathcal{L}^k - \operatorname{fold}_i(M_i^k + \mu Q_i^k) \right) + \mathcal{T}^k - \mu \mathcal{Y}^k \Big).$$

- For term $\mathcal{E}^{k+1}$:
$$\mathcal{E}^{k+1} = D_{\gamma \mu \tau_1} \left( \mathcal{E}^k - \tau_1 \left( \mathcal{T}^k - \mu \mathcal{Y}^k \right) \right).$$

- For term $\Delta\tilde{\Gamma}^{k+1}$:
$$\Delta\tilde{\Gamma}^{k+1} = \Delta\tilde{\Gamma}^k - \tau_2 \left( \Delta\tilde{\Gamma}^k - \mathcal{L}^{k+1} - \mathcal{E}^{k+1} + \mathcal{A} \circ \Gamma + \mu \mathcal{Y}^k \right).$$

Here, $\Delta\tilde{\Gamma}^{k+1}$ is a tensor, we can transform it to its original form as follows.
$$\Delta\Gamma^{k+1} = \sum_{i=1}^n J_i^+ (\Delta\tilde{\Gamma}^{k+1})_{(3)}^\top \boldsymbol{\epsilon}_i \boldsymbol{\epsilon}_i^\top,$$
where $J_i^+ = (J_i^\top J_i)^{-1} J_i^\top$ is pseudo-inverse of $J_i$ and $(\Delta\tilde{\Gamma}^{k+1})_{(3)}$ is the mode-3 unfolding matrix of tensor $\Delta\tilde{\Gamma}^{k+1}$.

- For terms $\mathcal{Y}^{k+1}$ and $Q_i^{k+1}$:
$$\mathcal{Y}^{k+1} = \mathcal{Y}^k - \mathcal{T}^{k+1}/\mu; \quad Q_i^{k+1} = Q_i^k - (\mathcal{L}_{(i)}^{k+1} - M_i^{k+1})/\mu, \quad i = 1, 2, 3.$$

The global convergence of the above optimization process is guaranteed by the following theorem.

**Theorem 1** *The sequence $\{M_i^k, \mathcal{L}^k, \mathcal{E}^k, \Delta\tilde{\Gamma}^k, \mathcal{Y}^k, Q_i^k, \ i = 1, 2, 3\}$ generated by the above proximal gradient descent scheme with $\tau_1 < 1/5$ and $\tau_2 < 1$ converges to the optimal solution to Problem (10).*

*Proof.* The proof of convergence can be found in the supplementary material.

As we see in Eq. (10), the optimization problem is similar to the problems addressed in [6, 1]. However, the proposed work differs from these earlier work in the following respects:

1. RASL and TILT can be viewed as two special cases of our work. Consider the mode-3 unfolding matrix $A_{(3)}$ in the bottom row of Fig. 1. Suppose the tensor is formed by stacking a set of images along the third mode. Setting $\alpha_1 = 0$, $\alpha_2 = 0$ and $\alpha_3 = 1$, our method reduces to RASL. While for the mode-1 and mode-2 unfolding matrices (see Fig. 1), if we set $\alpha_1 = 0.5$, $\alpha_2 = 0.5$ and $\alpha_3 = 0$, our method reduces to TILT. In this sense, our formulation is more general as it tends to simultaneously perform rectification and alignment.

2. *Our work vs. RASL:* In the image alignment applications, RASL treats each image as a vector and does not make use of any spatial structure within each image. In contrast, as shown in Fig. 1, in our work, the low-rank constraint on the mode-1 and mode-2 unfolding matrices effectively harnesses the spatial structures within images.

3. *Our work vs. TILT:* TILT deals with only one image and harnesses spatial low-rank structures to rectify the image. However, TILT ignores the temporal correlation among multiple images. Our work combines the merits of RASL and TILT, and thus can extract more structural information in the visual data.

## 4 Experimental Results

In this section, we compare the proposed algorithm with two algorithms: RASL [6] and Li's work [14] (TILT [1] is not adopted for comparison because it can deal with only one sample). We choose them for comparison because: (1) They represent the latest work that address similar problems as ours. (2) The effectiveness and efficiency of our optimization method for recovery of low-rank tensors can be validated by comparing our work with RASL and Li's work; These algorithms are tested with several synthetic and real-world datasets, and the results are both qualitatively and quantitatively analyzed.

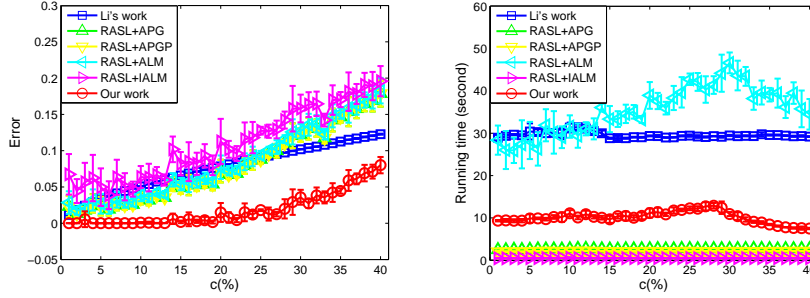

Figure 2: Results on synthetic data.

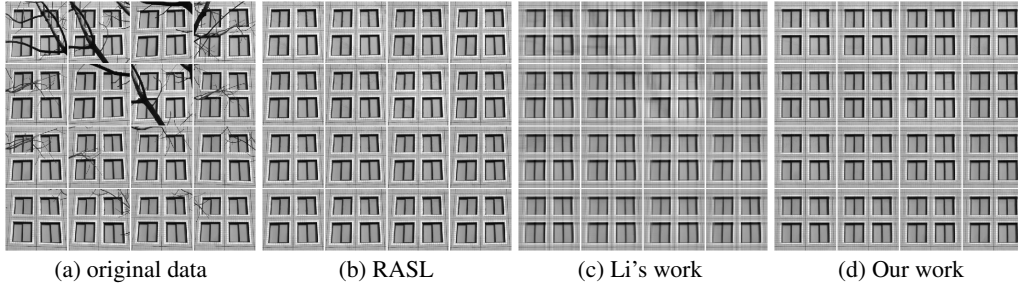

| (a) original data | (b) RASL | (c) Li's work | (d) Our work |

Figure 3: Results on the first data set.

**Results on Synthetic Data.** This part tests the above three algorithms with synthetic data. To make a fair comparison, some implementation details are clarified as follows: (1) Since domain transformations are not considered in Li's work, we assume the synthetic data are well aligned. (2) To eliminate the influence of different optimization methods, RASL is implemented with the following four optimization methods: APG (Accelerated Proximal Gradient), APGP (Accelerated Proximal Gradient with partial SVDs), ALM (Augmented Lagrange Multiplier) and IALM (Inexact Augmented Lagrange Multiplier)[2]. Moreover, since RASL is applied to one mode of the tensor, to make it more competitive, we apply RASL to each mode of the tensor and take the mode that has the minimal reconstruction error.

For synthetic data, we first randomly generate two data tensors: (1) a pure low-rank tensor $\mathcal{L}_o \in \mathbb{R}^{50 \times 50 \times 50}$ whose rank is (10,10,10); (2) an error tensor $\mathcal{E} \in \mathbb{R}^{50 \times 50 \times 50}$ in which only a fraction $c$ of entries are non-zero (To ensure the error to be sparse, the maximal value of $c$ is set to 40%). Then the testing tensor $\mathcal{A}$ can be obtained as $\mathcal{A} = \mathcal{L}_o + \mathcal{E}$. All the above three algorithms are applied to recover the low-rank structure of $\mathcal{A}$, which is represented as $\mathcal{L}_r$. Therefore, the reconstruction error is defined as $\text{error} = \frac{||\mathcal{L}_o - \mathcal{L}_r||_F}{||\mathcal{L}_o||_F}$. The result of a single run is a random variable, because the data are randomly generated, so the experiment is repeated 50 times to generate statistical averages.

The left column of Fig. 2 shows the reconstruction error, from which we can see that our work can achieve the most accurate result of reconstruction among all the algorithms. Even when 40% of entries are corrupted, the reconstruction error of our work is about 0.08. As shown in right column of Fig. 2, comparing with Li's work and RASL+ALM, our work can achieve about 3-4 times speed-up. Moveover, the result shows that the average running time of our work is higher than RASL+APG, RASL+APGP and RASL+IALM. However, these three methods only optimize on a single mode while our work optimize on all three modes and the variables evolved in (10) are about three times of those in RASL. The above results demonstrate the effectiveness and efficiency of our proposed optimization method for low-rank tensor recovery.

**Results on Real World Data.** In this part, we apply all three algorithms (RASL here is solved by ALM which gives the best results) to several real-world datasets. The first dataset contains 16 images of the side of a building, taken from various viewpoints by a perspective camera, and with various occlusions due to tree branches. Fig. 3 illustrates the low-rank recovery results on this data set, in which Fig. 3(a) shows the original image and Fig. 3(b)-(d) show the results of the three algorithms. Compared with RASL, we can see that our work and Li's work not only remove the

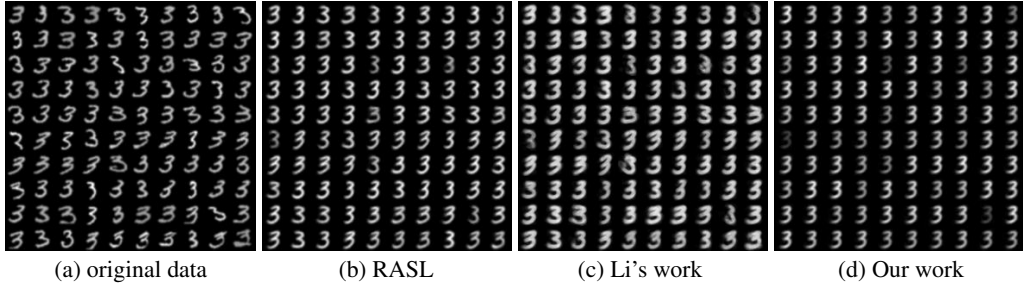

| (a) original data | (b) RASL | (c) Li's work | (d) Our work |

Figure 4: Results on the second data set.

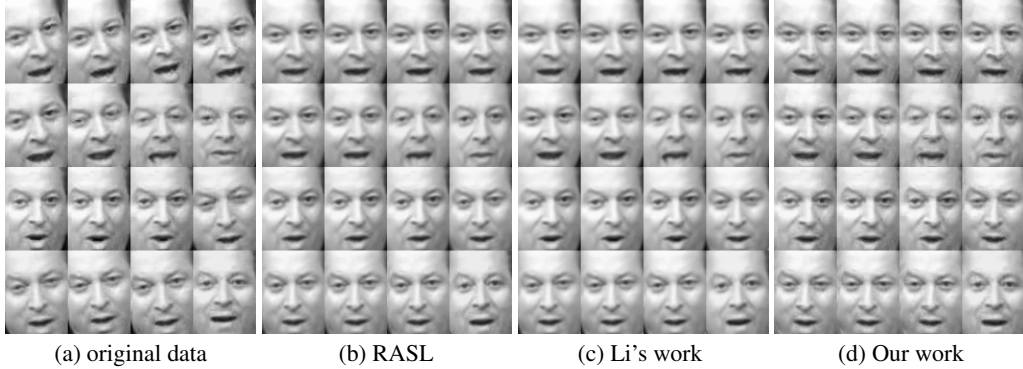

| (a) original data | (b) RASL | (c) Li's work | (d) Our work |

Figure 5: Results on the third data set.

branches from the windows, but also rectifiy window position. Moreover, the result obtained by our work is noticeably sharper than Li's work.

The second data set contains 100 images of the handwritten number "3", with a fair amount of diversity. For example, as shown in Fig. 4(a), the number "3" in the column 1 and row 6 is barely recognizable. The results of the three algorithms on this dataset are shown in Fig. 4(b)-(d). We can see that our work has achieved better performance than the other two algorithms from human's perception, in which the 3's are more clear and their poses are upright.

The third data set contains 140 frames of a video showing Al Gore talking. As shown in Fig. 5, the face alignment results obtained by our work is significantly better than those obtained by the other two algorithms. The reason is that human face has a rich spatial low-rank structures due to symmetry, and our method simultaneously harnesses both temporal and spatial low-rank structures for rectification and alignment.

# 5   Conclusion

We have in this paper proposed a general low-rank recovery framework for arbitrary tensor data, which can simultaneously realize rectification and alignment. We have adopted a proximal gradient based alternating direction method to solve the optimization problem, and have shown that the convergence of our algorithm is guaranteed. By comparing our work with the three state-of-the-art work through extensive simulations, we have demonstrated the effectiveness and efficiency of our method.

# 6   Acknowledgment

This work is partly supported by NSFC (Grant Nos. 61100147, 61203241 and 61305035), Zhejiang Provincial Natural Science Foundation (Grants Nos. LY12F03016, LQ12F03004 and LQ13F030009).

## Footnotes

[1]In most applications, a three-order tensor can be naturally partitioned into a set of matrices (such as image frames in a video) and transformations should be applied on these matrices

[2]For more detail, please refer to http://perception.csl.illinois.edu/matrix-rank/sample_code.html

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
