[Supplementary Material]

# Supplementary Material

**Xiaoqin Zhang, Di Wang**
Institute of Intelligent System and Decision
Wenzhou University
zhangxiaoqinnan@gmail.com, wangdi@wzu.edu.cn

**Zhengyuan Zhou**
Department of Electrical Engineering
Stanford University
zyzhou@stanford.edu

**Yi Ma**
Visual computing group
Microsoft Research Asia
mayi@microsoft.com

## 1  Proof of convergence

In this section, we study the global convergence of the proposed algorithm for solving (1).

$$\min_{\mathcal{L},\mathcal{E},M_i,\Delta\tilde{\Gamma}} \quad \sum_{i=1}^{3} \alpha_i||M_i||_* + \gamma||\mathcal{E}||_1$$
$$\text{s.t.} \quad \mathcal{A} \circ \Gamma + \Delta\tilde{\Gamma} = \mathcal{L} + \mathcal{E}$$
$$L_{(i)} = M_i, \quad i = 1, 2, 3 \tag{1}$$

If all matrices and tensors are column-vectorized by taking their columns and stacking them on one another to form column vectors, and let $\boldsymbol{m}_i$, $\boldsymbol{l}$, $\boldsymbol{e}$, $\Delta\tilde{\boldsymbol{\tau}}$, $\boldsymbol{q}_i$, $\boldsymbol{y}$ and $\boldsymbol{a}_0$ be the column vectors of $M_i$, $\mathcal{L}$, $\mathcal{E}$, $\Delta\tilde{\boldsymbol{\Gamma}}$, $Q_i$, $\mathcal{Y}$ and $\mathcal{A} \circ \Gamma$ respectively. Let $\boldsymbol{x} = \begin{pmatrix} \boldsymbol{m}_1 \\ \boldsymbol{m}_2 \\ \boldsymbol{m}_3 \\ \boldsymbol{l} \\ \boldsymbol{e} \end{pmatrix}$, $\boldsymbol{\eta} = \begin{pmatrix} \boldsymbol{y} \\ \boldsymbol{q}_1 \\ \boldsymbol{q}_2 \\ \boldsymbol{q}_3 \end{pmatrix}$, $A =$

$\begin{pmatrix} 0 & 0 & 0 & I & I \\ -I & 0 & 0 & P_1 & 0 \\ 0 & -I & 0 & P_2 & 0 \\ 0 & 0 & -I & P_3 & 0 \end{pmatrix}$, $B = \begin{pmatrix} -I \\ 0 \\ 0 \\ 0 \end{pmatrix}$, and $\boldsymbol{d} = \begin{pmatrix} \boldsymbol{a}_0 \\ 0 \\ 0 \\ 0 \end{pmatrix}$, where $P_1$, $P_2$ and $P_3$ are

permutation matrices. Then the optimization problem (1) becomes

$$\min_{\boldsymbol{x},\Delta\tilde{\boldsymbol{\tau}}} \quad f(\boldsymbol{x})$$
$$\text{s.t.} \quad A\boldsymbol{x} + B\Delta\tilde{\boldsymbol{\tau}} = \boldsymbol{d}, \tag{2}$$

where $f(\boldsymbol{x})$ is vector form of $\sum_{i=1}^{3} \alpha_i||M_i||_* + \gamma||\mathcal{E}||_1$.

Accordingly, the proximal gradient descent scheme in Section 3.2 can be written as

$$\begin{cases} x^{k+1} := \arg\min_{\boldsymbol{x}} \; f(\boldsymbol{x}) + \frac{1}{2\mu\tau_1} \left\| \boldsymbol{x} - \left( \boldsymbol{x}^k - \tau_1 A^\top (A\boldsymbol{x}^k + B\Delta\tilde{\boldsymbol{\tau}}^k - \boldsymbol{d} - \mu\boldsymbol{\eta}^k) \right) \right\|^2 & (3) \\[2ex] \Delta\tilde{\boldsymbol{\tau}}^{k+1} := \arg\min_{\Delta\tilde{\boldsymbol{\tau}}} \; \frac{1}{2\mu\tau_2} \left\| \Delta\tilde{\boldsymbol{\tau}} - \left( \Delta\tilde{\boldsymbol{\tau}}^k - \tau_2 B^\top \left( A\boldsymbol{x}^{k+1} + B\Delta\tilde{\boldsymbol{\tau}}^k - \boldsymbol{d} - \mu\boldsymbol{\eta}^k \right) \right) \right\|^2 & (4) \\[2ex] \boldsymbol{\eta}^{k+1} := \boldsymbol{\eta}^k - (A\boldsymbol{x}^{k+1} + B\Delta\tilde{\boldsymbol{\tau}}^{k+1} - \boldsymbol{d})/\mu & (5) \end{cases}$$

To prove the global convergence, we need to prove the following lemma.

**Lemma 1.** *Assume that $(\boldsymbol{x}^*, \Delta\tilde{\boldsymbol{\tau}}^*)$ is an optimal solution of (2) and $\boldsymbol{\eta}^*$ is the corresponding optimal Lagrange multipliers. If the step sizes $\tau_1 < 1/\lambda_{\max}(A^\top A)$ and $\tau_2 < 1/\lambda_{\max}(B^\top B)$, where $\lambda_{\max}(C)$ denotes the largest eigenvalue of matrix $C$. Then the sequence $(\boldsymbol{x}^k, \Delta\tilde{\boldsymbol{\tau}}^k, \boldsymbol{\eta}^k)$ generated by (3)-(5) satisfies:*

*(i) let $z^k = \frac{1}{\mu\tau_1}\|\boldsymbol{x}^k - \boldsymbol{x}^*\|^2 - \frac{1}{\mu}\|A(\boldsymbol{x}^k - \boldsymbol{x}^*)\|^2 + \frac{1}{\mu\tau_2}\|\Delta\tilde{\boldsymbol{\tau}}^k - \Delta\tilde{\boldsymbol{\tau}}^*\|^2 + \mu\|\boldsymbol{\eta}^k - \boldsymbol{\eta}^*\|^2$, the sequence $\{z^k\}_{k=1}^{\infty}$ is monotonically non-increasing;*

*(ii) $\|\boldsymbol{x}^{k+1} - \boldsymbol{x}^k\| \to 0$, $\|\Delta\tilde{\boldsymbol{\tau}}^{k+1} - \Delta\tilde{\boldsymbol{\tau}}^k\| \to 0$, and $\|\boldsymbol{\eta}^{k+1} - \boldsymbol{\eta}^k\| \to 0$.*

*Proof.* The optimality conditions for the subproblems (3) and (4) satisfy

$$-\frac{1}{\mu\tau_1}(\boldsymbol{x}^{k+1} - \boldsymbol{x}^k) - A^\top\left[\frac{1}{\mu}(A\boldsymbol{x}^k + B\Delta\tilde{\boldsymbol{\tau}}^k - \boldsymbol{d}) - \boldsymbol{\eta}^k\right] \in \partial f(\boldsymbol{x}^{k+1}) \tag{6}$$

$$(\Delta\tilde{\boldsymbol{\tau}}^{k+1} - \Delta\tilde{\boldsymbol{\tau}}^k) + \tau_2 B^\top(A\boldsymbol{x}^{k+1} + B\Delta\tilde{\boldsymbol{\tau}}^k - \boldsymbol{d} - \mu\boldsymbol{\eta}^k) = \boldsymbol{0} \tag{7}$$

$(\boldsymbol{x}^*, \Delta\tilde{\boldsymbol{\tau}}^*, \boldsymbol{\eta}^*)$ also satisfy the following equations from the KKT conditions

$$A^\top\boldsymbol{\eta}^* \in \partial f(\boldsymbol{x}^*) \tag{8}$$

$$B^\top\boldsymbol{\eta}^* = \boldsymbol{0} \tag{9}$$

$$A\boldsymbol{x}^* + B\Delta\tilde{\boldsymbol{\tau}}^* = \boldsymbol{d} \tag{10}$$

Note that the fact that $\partial f(\cdot)$ is a monotone operator, then

$$\langle \boldsymbol{x}^{k+1} - \boldsymbol{x}^*, \partial f(\boldsymbol{x}^{k+1}) - \partial f(\boldsymbol{x}^*)\rangle \geq 0 \tag{11}$$

Substitute (6) and (8) in (11) and use the updating formula (5), we have

$$
\begin{aligned}
0 \quad\leq\quad & \langle \boldsymbol{x}^{k+1} - \boldsymbol{x}^*, -\frac{1}{\mu\tau_1}(\boldsymbol{x}^{k+1} - \boldsymbol{x}^k) - A^\top\left[\frac{1}{\mu}(A\boldsymbol{x}^k + B\Delta\tilde{\boldsymbol{\tau}}^k - \boldsymbol{d}) - \boldsymbol{\eta}^k\right] - A^\top\boldsymbol{\eta}^*\rangle \\
=\quad & \langle \boldsymbol{x}^{k+1} - \boldsymbol{x}^*, -\frac{1}{\mu\tau_1}(\boldsymbol{x}^{k+1} - \boldsymbol{x}^k)\rangle - \langle A(\boldsymbol{x}^{k+1} - \boldsymbol{x}^*), \frac{1}{\mu}(A\boldsymbol{x}^k + B\Delta\tilde{\boldsymbol{\tau}}^k - \boldsymbol{d}) - \boldsymbol{\eta}^k + \boldsymbol{\eta}^*\rangle \\
=\quad & -\frac{1}{\mu\tau_1}\langle \boldsymbol{x}^{k+1} - \boldsymbol{x}^*, \boldsymbol{x}^{k+1} - \boldsymbol{x}^k\rangle + \langle A(\boldsymbol{x}^{k+1} - \boldsymbol{x}^*), \boldsymbol{\eta}^{k+1} - \boldsymbol{\eta}^*\rangle \\
& + \frac{1}{\mu}\langle A(\boldsymbol{x}^{k+1} - \boldsymbol{x}^*), A(\boldsymbol{x}^{k+1} - \boldsymbol{x}^k)\rangle + \frac{1}{\mu}\langle A(\boldsymbol{x}^{k+1} - \boldsymbol{x}^*), B(\Delta\tilde{\boldsymbol{\tau}}^{k+1} - \Delta\tilde{\boldsymbol{\tau}}^k)\rangle
\end{aligned} \tag{12}
$$

Multiply (7) by $\frac{1}{\mu\tau_2}$ and sum (9), we have

$$\frac{1}{\mu\tau_2}(\Delta\tilde{\boldsymbol{\tau}}^{k+1} - \Delta\tilde{\boldsymbol{\tau}}^k) + B^\top\left[\frac{1}{\mu}(A\boldsymbol{x}^{k+1} + B\Delta\tilde{\boldsymbol{\tau}}^k - \boldsymbol{d}) - \boldsymbol{\eta}^k + \boldsymbol{\eta}^*\right] = \boldsymbol{0} \tag{13}$$

Taking the inner product with $\Delta\tilde{\boldsymbol{\tau}}^{k+1} - \Delta\tilde{\boldsymbol{\tau}}^*$ on both sides of above equation and use the updating formula (5), we obtain

$$
\begin{aligned}
0 \quad=\quad & \frac{1}{\mu\tau_2}\langle \Delta\tilde{\boldsymbol{\tau}}^{k+1} - \Delta\tilde{\boldsymbol{\tau}}^*, \Delta\tilde{\boldsymbol{\tau}}^{k+1} - \Delta\tilde{\boldsymbol{\tau}}^k\rangle + \langle B(\Delta\tilde{\boldsymbol{\tau}}^{k+1} - \Delta\tilde{\boldsymbol{\tau}}^*), \boldsymbol{\eta}^* - \boldsymbol{\eta}^{k+1}\rangle \\
& - \frac{1}{\mu}\langle B(\Delta\tilde{\boldsymbol{\tau}}^{k+1} - \Delta\tilde{\boldsymbol{\tau}}^*), B(\Delta\tilde{\boldsymbol{\tau}}^{k+1} - \Delta\tilde{\boldsymbol{\tau}}^k)\rangle
\end{aligned} \tag{14}
$$

Using the KKT condition (10) and updating formula (5), (12) minus (14) is

$$
\begin{aligned}
0 \quad\leq\quad & -\frac{1}{\mu\tau_1}\langle \boldsymbol{x}^{k+1} - \boldsymbol{x}^*, \boldsymbol{x}^{k+1} - \boldsymbol{x}^k\rangle - \frac{1}{\mu\tau_2}\langle \Delta\tilde{\boldsymbol{\tau}}^{k+1} - \Delta\tilde{\boldsymbol{\tau}}^*, \Delta\tilde{\boldsymbol{\tau}}^{k+1} - \Delta\tilde{\boldsymbol{\tau}}^k\rangle \\
& - \mu\langle \boldsymbol{\eta}^{k+1} - \boldsymbol{\eta}^k, \boldsymbol{\eta}^{k+1} - \boldsymbol{\eta}^*\rangle - \langle \boldsymbol{\eta}^{k+1} - \boldsymbol{\eta}^k, B(\Delta\tilde{\boldsymbol{\tau}}^{k+1} - \Delta\tilde{\boldsymbol{\tau}}^k)\rangle \\
& + \frac{1}{\mu}\langle A(\boldsymbol{x}^{k+1} - \boldsymbol{x}^*), A(\boldsymbol{x}^{k+1} - \boldsymbol{x}^k)\rangle
\end{aligned}
$$

By using the fact

$$2\langle \boldsymbol{a}^{k+1} - \boldsymbol{a}^*, \boldsymbol{a}^{k+1} - \boldsymbol{a}^k\rangle = \|\boldsymbol{a}^{k+1} - \boldsymbol{a}^*\|^2 - \|\boldsymbol{a}^k - \boldsymbol{a}^*\|^2 + \|\boldsymbol{a}^{k+1} - \boldsymbol{a}^k\|^2$$

we obtain

$$z^{k+1} \leq z^k - \left[ \frac{1}{\mu \tau_1} \|\boldsymbol{x}^{k+1} - \boldsymbol{x}^k\|^2 - \frac{1}{\mu} \|A(\boldsymbol{x}^{k+1} - \boldsymbol{x}^k)\|^2 \right]$$

$$- \left[ \frac{1}{\mu \tau_2} \|\Delta \tilde{\boldsymbol{\tau}}^{k+1} - \Delta \tilde{\boldsymbol{\tau}}^k\|^2 + \mu \|\boldsymbol{\eta}^{k+1} - \boldsymbol{\eta}^k\|^2 + 2\langle \boldsymbol{\eta}^{k+1} - \boldsymbol{\eta}^k, B(\Delta \tilde{\boldsymbol{\tau}}^{k+1} - \Delta \tilde{\boldsymbol{\tau}}^k)\rangle \right] \quad (15)$$

On the other hand, since $\tau_1 < 1/\lambda_{\max}(A^\top A)$ and $\tau_2 < 1/\lambda_{\max}(B^\top B)$, the following equations hold:

$$\frac{1}{\tau_1} \|\boldsymbol{x}^{k+1} - \boldsymbol{x}^k\|^2 - \|A(\boldsymbol{x}^{k+1} - \boldsymbol{x}^k)\|^2 \geq \left( \frac{1}{\tau_1} - \lambda_{\max}(A^\top A) \right) \|\boldsymbol{x}^{k+1} - \boldsymbol{x}^k\|^2 \geq 0 \quad (16)$$

$$\frac{1}{\mu \tau_2} \|\Delta \tilde{\boldsymbol{\tau}}^{k+1} - \Delta \tilde{\boldsymbol{\tau}}^k\|^2 + \mu \|\boldsymbol{\eta}^{k+1} - \boldsymbol{\eta}^k\|^2 + 2\langle \boldsymbol{\eta}^{k+1} - \boldsymbol{\eta}^k, B(\Delta \tilde{\boldsymbol{\tau}}^{k+1} - \Delta \tilde{\boldsymbol{\tau}}^k)\rangle$$

$$\geq \quad \frac{1}{\mu \tau_2} \|\Delta \tilde{\boldsymbol{\tau}}^{k+1} - \Delta \tilde{\boldsymbol{\tau}}^k\|^2 + \mu \|\boldsymbol{\eta}^{k+1} - \boldsymbol{\eta}^k\|^2 - \left[ \mu \|\boldsymbol{\eta}^{k+1} - \boldsymbol{\eta}^k\|^2 + \frac{1}{\mu} \|B(\Delta \tilde{\boldsymbol{\tau}}^{k+1} - \Delta \tilde{\boldsymbol{\tau}}^k)\|^2 \right]$$

$$= \quad \frac{1}{\mu \tau_2} \|\Delta \tilde{\boldsymbol{\tau}}^{k+1} - \Delta \tilde{\boldsymbol{\tau}}^k\|^2 - \frac{1}{\mu} \|B(\Delta \tilde{\boldsymbol{\tau}}^{k+1} - \Delta \tilde{\boldsymbol{\tau}}^k)\|^2$$

$$\geq \quad \frac{1}{\mu} \left( \frac{1}{\tau_2} - \lambda_{\max}(B^\top B) \right) \|\Delta \tilde{\boldsymbol{\tau}}^{k+1} - \Delta \tilde{\boldsymbol{\tau}}^k\|^2 \geq 0 \quad (17)$$

Combining (15), (16) and (17), the proof of (i) is complete.

As the sequence $\{z_k\}_{k=1}^\infty$ is monotonically non-increasing and non-negative, it has a limit. Then from (15), we can see that

$$\frac{1}{\tau_1} \|\boldsymbol{x}^{k+1} - \boldsymbol{x}^k\|^2 - \|A(\boldsymbol{x}^{k+1} - \boldsymbol{x}^k)\|^2 \to 0$$

$$\frac{1}{\mu \tau_2} \|\Delta \tilde{\boldsymbol{\tau}}^{k+1} - \Delta \tilde{\boldsymbol{\tau}}^k\|^2 + \mu \|\boldsymbol{\eta}^{k+1} - \boldsymbol{\eta}^k\|^2 + 2\langle \boldsymbol{\eta}^{k+1} - \boldsymbol{\eta}^k, B(\Delta \tilde{\boldsymbol{\tau}}^{k+1} - \Delta \tilde{\boldsymbol{\tau}}^k)\rangle \to 0$$

Combining (16) and (17), we have

$$\|\boldsymbol{x}^{k+1} - \boldsymbol{x}^k\| \to 0, \|\Delta \tilde{\boldsymbol{\tau}}^{k+1} - \Delta \tilde{\boldsymbol{\tau}}^k\| \to 0$$

Note that

$$\frac{1}{\mu \tau_2} \|\Delta \tilde{\boldsymbol{\tau}}^{k+1} - \Delta \tilde{\boldsymbol{\tau}}^k\|^2 + \mu \|\boldsymbol{\eta}^{k+1} - \boldsymbol{\eta}^k\|^2 + 2\langle \boldsymbol{\eta}^{k+1} - \boldsymbol{\eta}^k, B(\Delta \tilde{\boldsymbol{\tau}}^{k+1} - \Delta \tilde{\boldsymbol{\tau}}^k)\rangle$$

$$= \quad \frac{1}{\mu \tau_2} \|\Delta \tilde{\boldsymbol{\tau}}^{k+1} - \Delta \tilde{\boldsymbol{\tau}}^k\|^2 + \mu \|\boldsymbol{\eta}^{k+1} - \boldsymbol{\eta}^k\|^2 + 2\langle B^\top(\boldsymbol{\eta}^{k+1} - \boldsymbol{\eta}^k), \Delta \tilde{\boldsymbol{\tau}}^{k+1} - \Delta \tilde{\boldsymbol{\tau}}^k\rangle$$

$$\geq \quad \frac{1}{\mu \tau_2} \|\Delta \tilde{\boldsymbol{\tau}}^{k+1} - \Delta \tilde{\boldsymbol{\tau}}^k\|^2 + \mu \|\boldsymbol{\eta}^{k+1} - \boldsymbol{\eta}^k\|^2 - \left[ \frac{1}{\mu \tau_2} \|\Delta \tilde{\boldsymbol{\tau}}^{k+1} - \Delta \tilde{\boldsymbol{\tau}}^k\|^2 + \mu \tau_2 \|B^\top(\boldsymbol{\eta}^{k+1} - \boldsymbol{\eta}^k)\|^2 \right]$$

$$\geq \quad \mu \tau_2 \left( \frac{1}{\tau_2} - \lambda_{\max}(B^\top B) \right) \|\boldsymbol{\eta}^{k+1} - \boldsymbol{\eta}^k\|^2$$

Thus $\|\boldsymbol{\eta}^{k+1} - \boldsymbol{\eta}^k\| \to 0$. The proof of (ii) is complete. $\qquad \square$

In the following, we will give the global convergence result of our proposed algorithm.

**Theorem 1.** *The sequence $\{(\boldsymbol{x}^k, \Delta \tilde{\boldsymbol{\tau}}^k, \boldsymbol{\eta}^k)\}_{k=1}^\infty$ generated by Section 3.2 with $\tau_1 < 1/5$ and $\tau_2 < 1$ converges to an optimal solution to problem (1).*

*Proof.* It is easy to check that $\lambda_{\max}(A^\top A) = 1$ and $\lambda_{\max}(B^\top B) = 5$.

From Lemma 1(i), we know the sequence $\{(\boldsymbol{x}^k, \Delta \tilde{\boldsymbol{\tau}}^k, \boldsymbol{\eta}^k)\}_{k=1}^\infty$ is bounded. Hence, there exists a convergent subsequence such that $\lim_{j \to \infty} (\boldsymbol{x}^{k_j}, \Delta \tilde{\boldsymbol{\tau}}^{k_j}, \boldsymbol{\eta}^{k_j}) = (\boldsymbol{x}_0, \Delta \tilde{\boldsymbol{\tau}}_0, \boldsymbol{\eta}_0)$. Then from the updating formula (5) and Lemma 1(ii), we have

$$A\boldsymbol{x}_0 + B\Delta \tilde{\boldsymbol{\tau}}_0 - \boldsymbol{d} = 0 \quad (18)$$

By letting $k = k_j - 1$ in (6) and $k = k_j$ in (7), we have

$$-\frac{1}{\mu\tau_1}(\boldsymbol{x}^{k_j} - \boldsymbol{x}^{k_j-1}) - A^\top \left\{ \frac{1}{\mu}(A\boldsymbol{x}^{k_j} + B\Delta\tilde{\boldsymbol{\tau}}^{k_j} - \boldsymbol{d}) - \boldsymbol{\eta}^{k_j} + \right.$$

$$\left. \frac{1}{\mu}\left[A(\boldsymbol{x}^{k_j-1} - \boldsymbol{x}^{k_j}) + B(\Delta\tilde{\boldsymbol{\tau}}^{k_j-1} - \Delta\tilde{\boldsymbol{\tau}}^{k_j})\right] + \boldsymbol{\eta}^{k_j} - \boldsymbol{\eta}^{k_j-1} \right\} \in \partial f(\boldsymbol{x}^{k_j})$$

$$(\Delta\tilde{\boldsymbol{\tau}}^{k_j+1} - \Delta\tilde{\boldsymbol{\tau}}^{k_j}) + \tau_2 B^\top (A\boldsymbol{x}^{k_j} + B\Delta\tilde{\boldsymbol{\tau}}^{k_j} - \boldsymbol{d} - \mu\boldsymbol{\eta}^{k_j} + A(\boldsymbol{x}^{k_j+1} - \boldsymbol{x}^{k_j})) = \boldsymbol{0}$$

Let $j \to \infty$, by Lemma 1(ii), we have

$$A^\top \boldsymbol{\eta}_0 \in \partial f(\boldsymbol{x}_0) \tag{19}$$

$$B^\top \boldsymbol{\eta}_0 = \boldsymbol{0} \tag{20}$$

(18), (19) and (20) show that $(\boldsymbol{x}_0, \Delta\tilde{\boldsymbol{\tau}}_0, \boldsymbol{\eta}_0)$ satisfies the KKT conditions for (2) and thus is an optimal solution to (2).

To complete the proof, we next show that the whole sequence $\{(\boldsymbol{x}^k, \Delta\tilde{\boldsymbol{\tau}}^k, \boldsymbol{\eta}^k)\}_{k=1}^\infty$ converges to $(\boldsymbol{x}_0, \Delta\tilde{\boldsymbol{\tau}}_0, \boldsymbol{\eta}_0)$. By choosing $(\boldsymbol{x}^*, \Delta\tilde{\boldsymbol{\tau}}^*, \boldsymbol{\eta}^*) = (\boldsymbol{x}_0, \Delta\tilde{\boldsymbol{\tau}}_0, \boldsymbol{\eta}_0)$ in Lemma 1 (i), we have

$$\frac{1}{\mu\tau_1}\|\boldsymbol{x}^{k_j} - \boldsymbol{x}_0\|^2 - \frac{1}{\mu}\|A(\boldsymbol{x}^{k_j} - \boldsymbol{x}_0)\|^2 + \frac{1}{\mu\tau_2}\|\Delta\tilde{\boldsymbol{\tau}}^{k_j} - \Delta\tilde{\boldsymbol{\tau}}_0\|^2 + \mu\|\boldsymbol{\eta}^{k_j} - \boldsymbol{\eta}_0\|^2 \to 0$$

By Lemma 1 (i), we know that the sequence $\{\frac{1}{\mu\tau_1}\|\boldsymbol{x}^k - \boldsymbol{x}_0\|^2 - \frac{1}{\mu}\|A(\boldsymbol{x}^k - \boldsymbol{x}_0)\|^2 + \frac{1}{\mu\tau_2}\|\Delta\tilde{\boldsymbol{\tau}}^k - \Delta\tilde{\boldsymbol{\tau}}_0\|^2 + \mu\|\boldsymbol{\eta}^k - \boldsymbol{\eta}_0\|^2\}_{k=1}^\infty$ has a limit. Hence,

$$\frac{1}{\mu\tau_1}\|\boldsymbol{x}^k - \boldsymbol{x}_0\|^2 - \frac{1}{\mu}\|A(\boldsymbol{x}^k - \boldsymbol{x}_0)\|^2 + \frac{1}{\mu\tau_2}\|\Delta\tilde{\boldsymbol{\tau}}^k - \Delta\tilde{\boldsymbol{\tau}}_0\|^2 + \mu\|\boldsymbol{\eta}^k - \boldsymbol{\eta}_0\|^2 \to 0$$

So $\lim_{k\to\infty}(\boldsymbol{x}^k, \Delta\tilde{\boldsymbol{\tau}}^k, \boldsymbol{\eta}^k) = (\boldsymbol{x}_0, \Delta\tilde{\boldsymbol{\tau}}_0, \boldsymbol{\eta}_0)$. The proof is complete. $\qquad\square$