[Reviews · NeurIPS 2013]

Submitted by Assigned_Reviewer_2

The authors present an unnamed algorithm for recovering a degree three tensor from a noisy version, where the noise is due to (a) slice misalignment and (b) sparse noise. The authors present a loss function for the model which they optimize by an ADMM-inspired gradient descent heuristic. The authors compare their method on real and synthetic image data to different implementations of RASL and and an algorithm from [14] which they call "Li's work" and show that their algorithm has lower recovery error.

The paper is clearly written and the idea of performing alignment and denoising on multiple images at once seems to be novel, while the reviewer is not a full expert on tensor methods in image processing and cannot finally settle the question of originality of the application. The heuristic used in the optimization method is somewhat original but can be obtained by putting known methods together.

It surprises the reviewer that theorem 1 on global convergence claims to be true without further assumptions on the noise model or the underlying tensor. Reading through the appendix reveals that there are indeed conditions depending on those. Maybe the authors can clarify this.

Also, the authors mention TILT which could be used slice-wise. Maybe as a sanity check, a comparison to slice-wise TILT could be added to the experiments.

The significance of the algorithm relies in my opinion not on the theoretical groundwork which is only partly new, but mainly on the practical applications, in which it seems to perform significantly better than standard methods.
Summary: A novel optimization algorithm for simultaneously rectifying and denoising of degree three tensors which seems to outperform state-of-the-art methods. In this combination, it seems to be novel and relevant to the NIPS community.

Submitted by Assigned_Reviewer_4

This paper proposes a model to simultaneously rectify and align images via robust low-rank tensor recovery. This model is a generalization of the corresponding matrix problem studied in TILT [1] and RASL [6], to the tensor correspondence. This work is actually a combination of a number of ideas in the existing literature. First, the model (6) is a generalization of the matrix problem in RASL. Second, relaxing the tensor rank to the sum of ranks of its unfoldings has been studied in [10]. Note that in Eq. (7), the L_{(i)} are unfoldings, thus this is not based on Tucker decomposition. The statement that is 5 lines before Eq. (7) is not appropriate.
Thrid, using ADMM to solve the resulted convex problem is not new. Similar idea has been studied in [6] and [11]. Thus, this paper is not of enough contribution in terms of novelty. The numerical results seem to be interesting though.
Summary: This paper is not of enough contribution in terms of novelty. The numerical results seem to be interesting though.

Submitted by Assigned_Reviewer_5

This paper is an extension of TILT [1] and RASL [6] that estimate alignments of images based on a tensor model for a collection of images.

Given a collection of images, the authors propose a method for finding a linear transformation that aligns each image simultaneously taking into account the low-rank-ness of the true image as a matrix and the correlation between different images. The authors formulate this problem as a low-rank tensor recovery problem with sparse corruption (similar to robust PCA).

An extension of robust PCA to tensor data was proposed by Li et al. [14] but they did not optimize the alignments.

Two previous studies that consider the same alignment model, TILT [1] and RASL [6] consider either the low-rank-ness of each image separately or the low-rank-ness of the collection of vectorized images, but do not consider the two jointly.

The authors derive a linearized ADM algorithm for the inner minimization problem and show its convergence. The authors also empirically demonstrate the advantage of joint low-rank modeling on several data sets.

The paper is mostly clearly written.

There are some issues I would like the authors to clarify:
1. If I undderstand correctly, the optimization algorithm discussed in Section 3.2 is an inner loop that only solves the linearized problem (9). However the outer loop is not described anyware on this paper. Please clarify.
2. Please quantitatively evaluate he results of Figures 3-5.
3. I only see sparse corruptions in Figure 3. How would using a squared loss as in [10-13] change the results of Figures 4-5?
Summary: This paper is a tensor extension of two previous studies that simultaneously considers two types of low-rank-ness in a collection of images. The paper is mostly clearly written and suitable for presentation.

Submitted by Assigned_Reviewer_6

This paper considers the (unrectified and misaligned) tensor recovery problem, which is an interesting problem.

Originality:
This work extends the matrix case in [1] to the tensor case. Specifically, it replaces the matrix nuclear norm in Eq. (3) of [1] by the tensor nuclear norm. I am afraid that this type of extension may not be treated as a substantial contribution any more since similar extensions have appeared in many early papers like [10-14, etc]. The new staff beyond this extension seems to consider the misalignment issue in tensors. The technique used to deal with unrectification and misalignment actually follows [1] as well. This paper just simply splits the tensor data into matrices and uses the technique proposed in [1] to deal with uncertification and misalignment in matrices. Although this paper claims that the proposed method align across images unlike [1], that is just due to the tensor nuclear norm rather than any new alignment technique.

A variant of the ADMM method is used to solve a key step of the proposed formulation. This paper claims ADMM is not efficient to solve (12-1) --- the first subproblem in (12) because (12-1) does not give a closed form. To my experiences, if properly introduce new variables into (11), it can be formulated into a structure good for ADMM (that is, all subproblems have closed form solutions): you can try duplicate \L and \E in (11).

The proof for Theorem 1 seems unnecessary. The convergence for many variants for ADMM, including using the proximal gradient descent (used in this paper) to replace the exactly solving, has been shown in many optimization papers already, for example, [A], [B].

Overall, this paper is not of sufficient contribution in term of originality. My novelty judgement is only based on the aspects of modeling and algorithm. Contributions in numerical results may be ignored.

Clarity:
Some equations are not clear enough, and some notations are abused apparently in section 3.1. Readers like me have to read [1] to understand those undefined and unclear notations and equations. Authors should seriously treat the following issues.
- (5) reuses "\circ" which has been used in (2). Two ``\circ'' actually mean totally different things. Readers that unfamiliar with [1] may be misled in (5). Moreover, the followed explanation for "A\circ \Tau" is also unclear. ``Applying the transformation \tau_i to each matrix A(:,:,i)'' sounds like the matrix multiplication \tau_i A(:,:,i). It actually means A(\tau_x, \tau_y, i) from [1].
- What is the value of ``p'' in two lines before (5)?
- (8) is suddenly relaxed into (9) by using a ``popular'' way. What is the ``popular'' way? No explanation, no citations. Readers have to figure out it from other papers like [1].
- The last line of Section 3.1: what is the value of "n". This notation "n" appears for multiple times before this line. Apparently, they denote different things.

[A] On the Linear Convergence of the Alternating Direction Method of Multipliers, Mingyi Hong, Zhi-Quan Luo, 2012.
[B] Linearized Alternating Direction Method of Multipliers for Constrained Linear Least-Squares Problem, Raymond H. Chan1, Min Tao, Xiaoming Yuan, 2012.
Summary: This paper is an incremental work from [1]. It is not of sufficient contribution in term of originality. My novelty judgement is only based on the aspects of modeling and algorithm. Contributions in numerical results may be ignored.
Author Feedback

Author rebuttal: 